# Language-guided Robot Grasping: CLIP-based Referring Grasp Synthesis in Clutter

**Georgios Tziafas**[1*]   **Yucheng Xu**[2*]   **Arushi Goel**[2]   **Mohammadreza Kasaei**[2]
**Zhibin Li**[3]   **Hamidreza Kasaei**[1]

[1]University of Groningen   [2]University of Edinburgh   [3]University College London

[1]{g.t.tziafas, h.kasaei}@rug.nl
[2]{Yucheng.Xu, A.Goel-1, m.kasaei}@ed.ac.uk
[3]alex.li@ucl.ac.uk

**Abstract:** Robots operating in human-centric environments require the integration of visual grounding and grasping capabilities to effectively manipulate objects based on user instructions. This work focuses on the task of *referring grasp synthesis*, which predicts a grasp pose for an object referred through natural language in cluttered scenes. Existing approaches often employ multi-stage pipelines that first segment the referred object and then propose a suitable grasp, and are evaluated in simple datasets or simulators that do not capture the complexity of natural indoor scenes. To address these limitations, we develop a challenging benchmark based on cluttered indoor scenes from OCID dataset, for which we generate referring expressions and connect them with 4-DoF grasp poses. Further, we propose a novel end-to-end model (CROG) that leverages the visual grounding capabilities of CLIP to learn grasp synthesis *directly* from image-text pairs. Our results show that vanilla integration of CLIP with pretrained models transfers poorly in our challenging benchmark, while CROG achieves significant improvements both in terms of grounding and grasping. Extensive robot experiments in both simulation and hardware demonstrate the effectiveness of our approach in challenging interactive object grasping scenarios that include clutter.

**Keywords:** Language-Guided Robot Grasping, Referring Grasp Synthesis, Visual Grounding

## 1   Introduction

Recent advancements in deep learning have paved the way for substantial breakthroughs in data-driven robotic grasping. Several works have proposed to synthesize grasps from purely visual inputs [1, 2, 3, 4]. In parallel, there is emerging work attempting to ground robotic perception [5, 6, 7] and action [8, 9] in natural language, aiming to enhance the ability of robots to interact with non-expert human users. In this work, we propose to bridge these two avenues via the task of *referring grasp synthesis*, where the robot is able to grasp a targeted object of interest that is indicated verbally by a human user (see Fig. 1). We focus on investigating this task in natural indoor scenes which include ambiguity and clutter, and are more realistic.

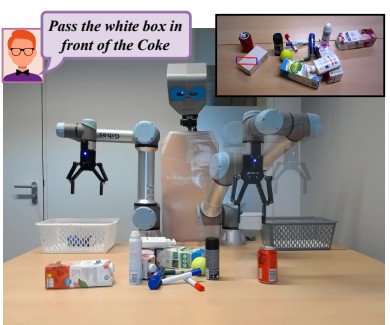

Figure 1: 4-DoF referring grasp synthesis in clutter.

Most existing approaches study interactive grasping scenarios via multi-stage pipelines [10, 11, 12, 13, 14], where first the target object is localized from a linguistic referring expression (i.e. *visual*

---

*Equal contribution

7th Conference on Robot Learning (CoRL 2023), Atlanta, USA.

*grounding*) and another module predicts a suitable grasp pose in a second step. The visual grounding models are trained either in benchmarks such as RefCOCO [15, 16, 17], which contain mostly outdoor scenes with few graspable objects, or custom datasets which limit the applicability of the learned model to fixed lab setups. Other robotics-related datasets collect language-annotated indoor scenes [5, 6, 7], but are not directed towards grasping, or contain grasp annotations but lack language [18, 19]. Additionally, related datasets that study clutter do not explicitly study ambiguous objects in the scene [7, 20], *i.e.*, objects of the same category appearing multiple times, and hence annotate only for such a category and a few attributes. They also mostly consider only pair-wise spatial relations between objects, which is not the case in free-form language (*e.g. "leftmost bowl"* is more natural than *"bowl left from other bowls"*).

Alternatively, a recent trend in language-based robot systems [21, 22] is to combine language models [23] with pretrained vision-language foundation models such as CLIP [24] for zero-shot grounding, and CLIP-based end-to-end grasping policies trained via imitation learning [25, 26]. Such approaches achieve impressive results but evaluate mostly in simulators which are fairly distant from natural realistic scenes, making the sim-to-real transfer more challenging.

To tackle the above limitations, we establish a new challenging dataset, *OCID-VLG*, that studies end-to-end vision-language-grasping in natural cluttered scenes. The dataset connects grasp annotations from the OCID-Grasp dataset [19] with referring expressions that include rich attribute vocabulary, model a broad range of relations, and explicitly consider ambiguity. Further, we propose an end-to-end model (**CLIP**-based **R**eferring **O**bject **G**rasping - *CROG*), that extends CLIP's visual grounding with both pixel-level segmentation, as in [27], and grasp synthesis tasks, via a novel multi-task objective. Experimental evaluations show that the proposed model is robust to referring expression complexity, and outperforms previous baselines that rely on vanilla integration of CLIP with the multi-stage approach. Extensive robot experiments demonstrate the effectiveness of the proposed model in challenging interactive grasping scenarios, in both simulation and real-world settings.

In summary, the main contributions of this work are: a) a new challenging dataset for visual grounding and referring grasp synthesis in cluttered scenes, comprising approximately 90 000 language-mask-grasp annotations, b) an end-to-end vision-language-grasping model, CROG, which efficiently learns a grasp policy by leveraging the powerful image-language alignment of CLIP, and demonstrate its performance merits compared to multi-stage baselines that utilize pretrained models, and, c) applying our proposed model in challenging interactive table cleaning scenarios through both simulation and real robot experiments.

## 2 Related Works

**Referring Expression Datasets**  Referring expressions are natural language descriptions that uniquely identify a target region in a paired image, often by referring to object attributes and spatial relations. Several datasets [28] have been proposed in the past, with expressions and target bounding boxes / masks annotated manually [29, 30] or via a two-player game [31]. Most popular benchmarks include Flickr30k-Entities [17] and the RefCOCO(/+,/g) suite [15, 32], containing annotations for MSCOCO [33] scenes, collected via the refer-it game strategy [31]. Alternatively, automatic referring expression generation is pursued via the usage of symbolic scene graph annotations and synthetic language templates [34, 35, 36, 16]. Above benchmarks concern referring expressions for RGB images with generic content and are mostly for outdoor scenes. Recent works propose datasets with referring expressions for objects in indoor environments and RGB-D / 3D visual data [7, 5, 6], but do not consider clutter and are not connected with robot grasping. In our work, we adopt the automatic generation method of CLEVR-Ref [36] to generate expressions for extracted scene graphs from OCID-Grasp dataset [19]. To the best of our knowledge, OCID-VLG is the first dataset that brings together referring expressions and grasp synthesis for cluttered indoor scenes.

**Visual Grounding**  Visual grounding is formulated in literature through the tasks of referring expression comprehension and referring image segmentation, depending on the type of localization required (box and mask respectively). Methods usually employ a two-stage detect-then-rank approach [37, 38, 39, 40], first generating object proposals and then ranking them according to their

correspondence to the expression. Single-stage methods [41, 42] alleviate the object proposal step by directly fusing vision-text features in a joint space. Recently, the Transformer architecture has been employed for both task variants separately [3, 43, 44], or jointly [45, 46], showcasing strong cross-modal alignment capabilities compared to previous CNN-LSTM fusion techniques. The grounding task has been recently adapted for 3D data [5], with similar two-stage methods fusing text features with point clouds [47, 48] or RGB-D views [49, 6]. Finally, transferring from large-scale vision-language pretraining [50, 51, 52, 24] is a common practice for usage in zero-shot [21, 22], or as an initialization for finetuning [25]. Similarly, in our work, we finetune the CLIP vision-language model [24] to further learn 4-DoF grasp synthesis in RGB.

**Grasp Synthesis** Grasp synthesis enables robots to determine the optimal way to grasp objects by considering visual information. Current grasp synthesis methods can be roughly categorized into 4-DoF and 6-DoF [53], according to the degrees of freedom (DoF) of the grasp configurations. **4-DoF** grasp synthesis [54, 19, 55] defines grasps by a 3D position and a top-down hand orientation ($yaw$), which is also commonly referred as a "*top-down grasp*". **6-DoF** grasp synthesis [18, 56, 57] defines grasp poses by 6D positions and orientations. Early works [1, 2] formulate 4-DoF grasp detection via decoding a set of grasp masks from the input RGB-D images and use camera calibration to transform the planar grasp into a gripper pose. Det-Seg [19] proposed a two-branch framework that generates semantic segmentation masks and uses them to refine the predicted grasps. SSG [58] introduced an instance-wise 4-DoF grasp synthesis framework and showed its effectiveness and robustness in cluttered scenarios. In this work, we build on the idea of using the segmentation mask as an extra signal for learning grasp synthesis, by making the masks object-specific via grounding them from referring expressions.

## 3 OCID-VLG Dataset

Visual grounding and grasp synthesis are mostly studied separately, and hence associated grounding datasets rarely involve cluttered indoor scenes [15, 17, 16] and lack grasp annotations [5, 6, 7], while grasp synthesis datasets lack language-grounding [55, 54, 18, 19]. Our proposed dataset, OCID-VLG (**V**ision-**L**anguage-**G**rasping), aims to cover this gap, by providing a joint dataset for both grounding and grasp synthesis in scenes from OCID dataset [59]. The dataset consists of 1 763 indoor tabletop RGB-D scenes with high clutter, including 31 object categories from a total of 58 unique instances. The OCID object catalog includes several object instances of the same category that vary in fine-grained details, granting it a desirable domain for integration with language. We manually annotate the catalog with a rich variety of object-related concepts, as well as pair-wise and absolute spatial relations. For each scene, we provide 2D segmentation masks and bounding boxes (at both category and instance level), as well as a complete parse of the scene, providing all 2D/3D locations, category, attribute and relation information for each object in a symbolic scene graph. We leverage the previous $75k$ hand-annotated 4-DoF grasp rectangles of OCID-Grasp [19] and connect each object in our scene graph with a set of grasp annotations. The grasp-annotated scene graphs are used to generate referring expressions with a custom variant of the CLEVR data engine [35]. We end up with $89,639$ unique scene-text-mask/box-grasp tuples, aimed to supervise both grounding and grasp synthesis tasks in an end-to-end fashion.

### 3.1 Referring Expression Generation

We first annotate a catalog of attribute and relation concepts that are used to refer to ambiguous objects in OCID-VLG scenes. For attributes, each object is annotated for its color, as well as with an instance-level description that refers to some object's property (e.g. brand, flavor, variety, maturity, function, texture, or shape). We note that not all objects are annotated for all mentioned concepts, but only for those that discriminate them from other objects of the same category. For spatial relations, we include both *relative* (e.g. *"bowl left from mug"*) as well as *absolute* location (e.g. *"leftmost bowl"*) concepts. We adapt the relation resolution heuristics of [44] and use the relation set {*"right", "rear right", "behind", "rear left", "left", "front left", "front", "front right", "on"*}, but augment it with the absolute location set {*"leftmost", "rightmost", "furthest", "closest"*}.

| Dataset | Clutter | Vision Data | Ref.Expr. Annot. | Grasp Annot. | Num.Obj. Categories | Num. Scenes | Num. Expr. | Parses |
|---|---|---|---|---|---|---|---|---|
| RefCOCO [15] | ✗ | RGB | box,mask | ✗ | 80 | 19.9k | 142.2k | ✗ |
| Flickr30k-Entities [17] | ✗ | RGB | box | ✗ | 44.5k | 31.7k | 158.9k | ✗ |
| CLEVR-Ref+ [36] | ✗ | RGB | box,mask | ✗ | - | 60k | 600k | ✔ |
| Cops-ref [16] | ✗ | RGB | box,mask | ✗ | 508 | 703 | 148.7k | ✔ |
| ScanRefer [5] | ✗ | 3D | box | ✗ | 250 | 800 | 51.5k | ✗ |
| ReferIt-RGBD [6] | ✗ | RGB-D | box | ✗ | - | 7.6k | 38.4k | ✗ |
| Sun-Spot [7] | ✔ | RGB-D | box | ✗ | 38 | 1.9k | 7.9k | ✗ |
| OCID-Ref [20] | ✔ | RGB,3D | box | ✗ | 58 | 2.3k | 305.6k | ✗ |
| Cornell [55] | ✗ | RGB-D | ✗ | 4-DoF | 240 | 1k | - | ✗ |
| Jacquard [54] | ✗ | RGB-D | ✗ | 4-DoF | - | 54k | - | ✗ |
| GraspNet [18] | ✔ | 3D | ✗ | 6-DoF | 88 | 190 | - | ✗ |
| OCID-Grasp [19] | ✔ | RGB-D | ✗ | 4-DoF | 31 | 1.7k | - | ✗ |
| OCID-VLG (ours) | ✔ | RGB-D,3D | box,mask | 4-DoF | 31 | 1.7k | 89.6k | ✔ |

Table 1: Comparison of main features and statistics between existing 2D / 3D visual grounding and grasp synthesis datasets with our OCID-VLG.

The complete list of the concept vocabulary and related statistics are provided in Appendix A.

After parsing scenes into scene graphs, we sample object and relation concepts to generate referring expressions using the CLEVR data engine [35] and custom templates, that follow the structure:

`[prefix] ([LOC`$_1$`] [ATT`$_1$`]) [OBJ`$_1$`] ((that is) [REL] the ([LOC`$_2$`] [ATT`$_2$`]) [OBJ`$_2$`])`

where `[OBJ]`, `[ATT]`, `[REL]`, `[LOC]` denote an object concept (category or instance-level), an attribute (color), a pair-wire relation and an absolute location respectively. The `[prefix]` is sampled from a set of general robot directives, e.g. *"Pick the"*. We construct template variations for 5 families, namely: a) **name**, (e.g. *"chocolate corn flakes"*), b) **attribute**, (e.g. *"brown cereal box package"*), c) **relation**, (e.g. *"corn flakes behind the bowl"*), d) **location**, (e.g. *"closest cereal box"*) and e) **mixed**, (e.g. *"cereal box to the rear left of the right apple"*), for a total of 56 distinct sub-templates. Note that templates (b)-(e) are constrained to only sample target objects that are ambiguous in the scene, hence attribute or relation information are needed to uniquely ground them.

## 3.2   Comparisons with Existing Datasets

OCID-VLG differs from existing datasets in several aspects and statistics, summarized in Table 1. Popular visual grounding datasets [15, 32, 17, 16] usually include few indoor scenes with cluttered content and provide only RGB data, limiting their applicability in the robotics domain. Robotics-related grounding datasets usually contain referring expressions for objects in room layouts [5, 6] (e.g. furniture), which are not directed towards grasping and do not consider clutter. Sun-Spot [7] contains tabletop cluttered scenes, but doesn't annotate segmentations and grasps and lacks absolute location annotations. Similarly, OCID-Ref [20] only provides boxes without segmentation and grasp annotations. We highlight that even though OCID-Ref could be used as a source of referring expressions for OCID-VLG, we chose to develop our own, as OCID-Ref expressions lack rich object and relation vocabulary, lack absolute relations and inherit corrupted labels from OCID dataset. Grasp synthesis datasets are either in object-level and not consider clutter [55, 54], or include cluttered scenes but do not annotate referring expressions [18, 19]. Additionally, most mentioned datasets do not explicitly consider ambiguities in the scene. In OCID-VLG, we use attributes and relations to refer to objects only in cases of ambiguity, aiming to prevent overfitting in superficial object-relation correspondences that exist in the training data. Finally, as we use the CLEVR [35] engine to generate expressions, our dataset is further equipped with symbolic parses of both the visual and the language modalities, which could be potentially utilized for training models with additional supervision.

## 4   Method

This section discusses the proposed task, the implemented baselines and the details of our end-to-end model, CROG.

**Problem Formulation.** Referring grasp synthesis considers the problems of referring image segmentation and grasp synthesis in tandem. Given an RGB image $I \in \mathbb{R}^{H \times W \times 3}$, a depth image $D \in \mathbb{R}^{H \times W}$ and a natural language expression $T$ that refers to a unique object in the scene, the goal is to predict both a pixel-wise segmentation mask of the referred object $M \in \{0, 1\}^{H \times W}$, as well as a grasp configuration $G = (x^*, y^*, \theta^*, l^*)$, where: $(W, H)$ the image resolution, $(x^*, y^*)$ the center of the optimal grasp in pixel coordinates, $\theta^*$ the gripper's rotation in camera reference frame and $l^*$ the gripper width in pixel coordinates. As in [1, 2, 19, 58], $G$ is recovered from three masks: a grasp scalar quality mask $Q \in \{0, 1\}^{H \times W}$, such that: $(x^*, y^*) = \texttt{argmax}_{(x,y)} Q(x, y)$, an angle mask $\Theta \in \{-\frac{\pi}{x}, \frac{\pi}{2}\}^{H \times W}$ and a width mask $L \in \{0, 1\}^{H \times W}$, such that: $\theta^* = \Theta(x^*, y^*)$, $l^* = L(x^*, y^*)$.

## 4.1 Multi-Stage Baselines

We design multi-stage baselines which integrate existing large-scale vision-language models with pretrained grasp synthesis models. To that end, we decompose the overall task into two stages, namely: a) a grounding function $f(I, T) = M$ that segments the referred object from the RGB image, and b) a grasp synthesis function $g(I, D, M) = G$ that isolates the object in the RGB-D inputs $I, D$ using the segmentation mask $M$ to produce a grasp $G$.

**Two-stage grounding with CLIP.** The grounding function can be further decomposed into two steps, first using an off-the-shell detector [60] for object proposal generation $f_{segm}(I) = \{M_n\}_{n=1}^N$, and then ranking the $N$ segmented object proposals according to their similarity with the language description $f_{rank}(M_n, T) = \texttt{argmax}_n S(M_n, T)$, where $S(\cdot)$ denotes a similarity metric between a segmented RGB object image $M_n$ and the language input $T$. In practise, we implement $S$ via CLIP's [24] pretraining objective, i.e. computing cosine similarity between visual features from passing $M_n$ to CLIP's visual encoder and a sentence-wide embedding of $T$ from CLIP's text encoder.

**Mask-conditioned grasp synthesis.** The grasp synthesis function is implemented via a pretrained network [1, 2] which receives the input pair $I, D$ and generates grasp $G$ via decoding the masks $Q, \Theta, L$. To isolate the desired object, given in mask $M$, we experimentally find that the best practise is to element-wise multiply the mask with the RGB-D inputs before passing to the network.

### 4.1.1 Zero-shot baselines

First, existing powerful pretrained models are experimented to assess their zero-shot performance in our challenging setup, including two multi-stage variants that use GR-ConvNet [1] pretrained on the Jacquard [54] dataset as grasp synthesis network, but differ in grounding as follows:

**SAM+CLIP.** In this setup, we use the Segment Anything (SAM) [60] model for instance segmentation and CLIP [24] for ranking as explained above. Similar to [61], we find that passing both a cropped box and the mask of the object to CLIP's feature extractor and ensembling the final similarities boosts performance.

**GLIP+SAM.** For this variant, we use a large pretrained visual grounding model, GLIP [62], for predicting a bounding box around the relevant object of interest given the natural language command, and prompt the SAM [60] model to get a tight segmentation mask for the object of interest.

### 4.1.2 Supervised baselines with vanilla CLIP integration

Second, we implemented a vanilla integration of CLIP in grasp synthesis models pretrained in OCID-Grasp [19], which aims to explore whether a model using segmentation and grasping trained on OCID scenes can be extended to our setup without any language conditioning. Similar to the SAM+CLIP setup, CLIP is used as a ranker and the supervised model as both a segmenter and grasper. We experiment with the two state-of-the-art models in OCID-Grasp, **Det-Seg** [19] and **SSG** [58], which can both provide both segmentation and grasp predictions for an input RGB-D scene.

## 4.2 CROG

We propose a model for estimating both the segmentation mask $M$ and the grasp masks $Q, \Theta, L$ in an end-to-end fashion (see the overview in Fig. 2). Our architecture is an extension of CRIS [27],

a model originally proposed for adapting CLIP to do pixel-level segmentation. CRIS achieves this with four main components: a) **unimodal encoders** for image and text, b) a **multi-modal FPN neck**, c) a **cross-attention vision-language decoder**, and d) projectors for **text-to-pixel contrastive loss**.

The unimodal encoders are initialized from CLIP's visual and text encoders [24], but utilize visual feature maps from 2th-4th stages of CLIP's ResNet-50 visual encoder $F_{v2} \in \mathbb{R}^{\frac{H}{8} \times \frac{W}{8} \times C_2}$, $F_{v3} \in \mathbb{R}^{\frac{H}{16} \times \frac{W}{16} \times C_3}$, $F_{v4} \in \mathbb{R}^{\frac{H}{32} \times \frac{W}{32} \times C_4}$, and considers both the sentence-level $F_s \in \mathbb{R}^{C'}$ and the token embeddings $F_t \in \mathbb{R}^{K \times C}$, where $C$ and $C'$ are the feature dimensions and $K$ the language sentence length.

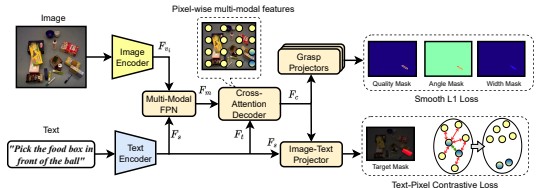

Figure 2: An overview of CROG.

Visual feature maps $F_{v2}, F_{v3}, F_{v4}$ and the sentence embedding $F_s$ are fused in feature pyramid style [63] via the multi-modal neck, in order to generate pixel-wise multi-modal representations $F_m \in \mathbb{R}^{N \times C}$ of the image-text pair, where $N = \frac{H}{16} \cdot \frac{W}{16}$.

The vision-language decoder uses a standard Transformer decoder [64] to let the multi-modal features $F_m$ cross-attend with all token embeddings $F_t$ and produce embeddings $F_c \in \mathbb{R}^{N \times C}$. This process adaptively propagates semantic information from text to visual features. Finally, $F_c$ and $F_s$ are projected into the same space, where contrastive alignment is performed via computing a binary cross entropy loss over the dot-product of the projected embeddings, pushing them together in the regions of the ground truth segmentation mask.

To obtain a mask prediction, the projected features $\hat{F}_c, \hat{F}_s$ are dot-producted with sigmoid activation, reshaped to $N = \frac{H}{4} \cdot \frac{W}{4}$ and upsampled to original image size. To adapt for grasp synthesis task, we propose to further add three projectors for generating grasp masks $Q, \Theta, L$ and supervise them with smooth L1-loss from the ground truth grasps, in parallel to the contrastive alignment loss of CRIS.

## 5 Experimental Results

This section evaluates our dataset using multi-stage baselines and compares them to our CROG model. Also, we conducted ablation studies to analyze the performance improvements and present the results of our robot experiments.

**Implementation** We initialize the vision and text encoders with the ResNet-50 and BERT weights from CLIP [24]. Input images are resized to $416 \times 416$, and texts are BPE-tokenized [64, 65]. The maximum length of input texts is set to 20. We train in multiple GPUs using the Adam optimizer with an initial learning rate $1e^{-4}$, that decays to $0.1$ over 35 epochs.

**Evaluation metrics** For grounding, we report *referring image segmentation (RIS)* [27, 66] metrics *IoU* and *Precision@X*. *IoU* calculates averaged intersection over union for the predicted segmentation and ground truth masks, while *Precision@X* measures the percentage of predictions with *IoU* higher than a threshold $X \in \{0.5, 0.6, 0.7, 0.8, 0.9\}$. For *referring grasp synthesis (RGS)*, Jacquard index *J@N* [54, 19, 58] is presented, measuring the percentage of top-N grasp predictions that have an angle difference within 30° and higher than 0.25 *IoU* with the ground truth grasp rectangle.

### 5.1 OCID-VLG Results

The grounding and grasp synthesis results are reported on the test split of OCID-VLG, containing $17.7k$ samples from held-out scenes of OCID. The test set contains seen objects but in novel scene configurations, resulting in unseen referring expressions. Results for zero-shot and supervised baselines and our CROG are in Table 2. Results show that baselines based on GR-ConvNet [1] pretrained on Jacquard [54], transfer poorly in OCID-VLG, even with ground truth grounding (28.7% J@1).

We find that the GR-ConvNet-based grasper tends to prefer edges, due to the top-down perspective of Jacquard images, which is not the case in OCID-VLG.

Zero-shot baselines score below 30% in both tasks, as due to the CLIP-based ranking methodology, *false positives* in grounding lead to incorrect grasping, regardless if the predicted grasp is correct for the mis-segmented object. Replacing large zero-shot models with supervised methods trained in OCID-Grasp (Det-Seg, SSG), offers a

| Method | RIS | | | | | | RGS | |
|---|---|---|---|---|---|---|---|---|
| | IoU | Pr@50 | Pr@60 | Pr@70 | Pr@80 | Pr@90 | J@1 | J@*Any* |
| GT-Grounding [†] | - | - | - | - | - | - | 28.7 | 70.2 |
| GT-Masks + CLIP [†] | 35.0 | 35.0 | 35.0 | 35.0 | 35.0 | 35.0 | 11.9 | 26.8 |
| SAM + CLIP [†] | 25.7 | 29.3 | 28.5 | 27.4 | 22.7 | 9.1 | 7.2 | 12.7 |
| GLIP + SAM [†] | 30.3 | 34.7 | 34.1 | 33.5 | 28.6 | 11.7 | 10.7 | 21.8 |
| Det-Seg + CLIP | 29.0 | 27.2 | 20.9 | 17.5 | 17.2 | 16.0 | 28.1 | 39.2 |
| SSG + CLIP | 33.6 | 35.6 | 35.6 | 35.5 | 35.5 | **32.8** | 33.5 | 34.7 |
| CROG (ours) | **81.1** | **96.9** | **94.8** | **87.2** | **64.1** | 16.4 | **77.2** | **87.7** |

Table 2: Comparison results in OCID-VLG test split. Baselines with [†] use GR-ConvNet [1] pretrained on Jacquard [54]. GT denotes the use of ground truth data for providing an upper bound of performance given perfect segmentation masks or grounding.

marginal improvement in grounding (+3.6% in IoU), but significant in grasping (+23.2% in J@1), while still low overall (39.2% J@*Any*). This indicates that even in presence of an OCID-specific grasper, the ranking methodology of vanilla CLIP integration significantly limits the grounding performance.

The proposed CROG overcomes such limitations by fine-tuning grounding and grasp synthesis together on top of CLIP, and surpasses previous methods with a large margin (+47.5% in IoU and +43.7% in J@1), offering a much more competitive baseline for the proposed OCID-VLG dataset.

## 5.2 Ablation Studies

Ablation studies are conducted to explore: a) the distribution of error according to referring expression type, and b) performance improvements of each main CROG component.

**Referring Expression Type** We first decompose the performance according to the type of the input expression and compare the analytical results of our model with the best-performing baseline, SSG+CLIP (see Fig. 3). We observe that the CLIP baseline struggles with grounding spatial concepts such as relations and locations (less than $30\%$ J@1), due to the loss of spatial information introduced by the segment-then-rank pipeline. On the contrary, CROG is trained with dense pixel-text token alignment via the cross-attention decoder, and is capable of spatial grounding and robust across all types.

**Effect of CROG components** We ablate the three main characteristics of CROG: a) initializing from CLIP, b) combining contrastive with grasp mask decoding tasks in a single objective, and c) dense text-pixel alignment with a decoder. Results are summarized in Table 3. All components are contributing to CROG's performance, where CLIP initialization is the most vital. Crucially, the removal of the contrastive or grasp loss results in a decrease in CROG's grasping or grounding capability. This highlights the knowledge transfer between the two tasks that justifies our selection of a multi-task training objective.

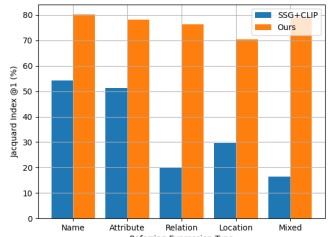

Figure 3: Grasp synthesis ablations according to the type of input referring expression.

| Method | RIS-*IoU* | RGS-*J@1* |
|---|---|---|
| CROG | **81.1** | **77.2** |
| - *w/o CLIP init* | 73.9 | 71.0 |
| - *w/o contrastive* | - | 73.4 |
| - *w/o grasp loss* | 79.3 | - |
| - *w/o decoder* | 78.2 | 72.3 |

Table 3: CROG ablation study.

## 5.3 Robot Experiments

We conducted experiments with a simulated and a real robot, where we want to evaluate the performance of our model in the context of an interactive table cleaning task. Our setup consists of a dual-arm robot with two UR5e manipulators with parallel jaw grippers and a Kinect sensor.

During each experiment, we randomly place 5-12 objects on a tabletop and provide language instruction to the robot to pick a target object and place it in a predefined container position.

We place objects in two scenarios, namely: a) **isolated**, where objects are scattered across the workspace, and b) **cluttered**, where we closely pack objects together. We note that in each scene we include distractor objects of the same category as the queried object. Our setup and example trials are shown in Fig. 4, while more qualitative results are provided in Appendix B.

| Setup | Fruit | Food Box | Food Can | Mug | Marker | Cereal | Flashlight | Overall |
|---|---|---|---|---|---|---|---|---|
| Isolated (#Trials) | 10 | 10 | 8 | 4 | 6 | 10 | 2 | 50 |
| Isolated - Ground.Acc. | 10 (100%) | 8 (80%) | 5 (63%) | 2 (50%) | 5 (83%) | 6 (60%) | 2 (100%) | 38 (76%) |
| Isolated - Succ.Rate | 10 (100%) | 5 (50%) | 4 (50%) | 1 (25%) | 5 (83%) | 4 (40%) | 2 (100%) | 31 (62%) |
| Cluttered (#Trials) | 10 | 10 | 8 | 4 | 6 | 10 | 2 | 50 |
| Cluttered - Ground.Acc. | 8 (80%) | 5 (50%) | 3 (38%) | 1 (25%) | 5 (83%) | 6 (60%) | 2 (100%) | 30 (60%) |
| Cluttered - Succ.Rate | 5 (50%) | 4 (40%) | 3 (38%) | 1 (25%) | 3 (50%) | 5 (50%) | 0 (0%) | 21 (42%) |

Table 4: Results of robot experiments Gazebo, where *Ground.Acc* denotes the number of trials where the target object is segmented correctly and *Succ.Rate* the number of successfully completed trials.

We conducted 50 trials per scenario in the Gazebo simulator [67], using object models from 7 categories of OCID-VLG, some as exact instances and others with different attributes. Object list, metrics, and recorded results are shown in Table 4. The robot achieves grounding accuracy of 76% (38/50) and success rate of 62% (31/50) in isolated and 60% (30/50), 42% (21/50) in cluttered scenes respectively, with grounding failures mostly for objects that are not similar in appearance to OCID-VLG categories. For real robot experiments, we initialize six unique scenes, three for isolated and three for cluttered scenarios, and provide grasping instructions for a total of 34 trials. We highlight that this test is more challenging, as the object set used for experiments has

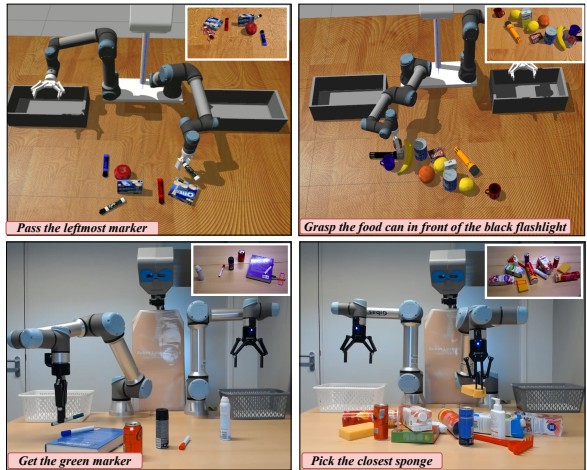

Figure 4: Interactive table cleaning trials in Gazebo *(top)* and real robot *(bottom)*, in isolated *(left column)* and cluttered *(right column)* scenes.

no overlap with OCID-VLG instances. In isolated scenes, the grounding accuracy is 65% and the success rate 23.9%, while in cluttered it is 60% and 20.0% respectively. In both experiments, the model is able to ground attribute concepts for unseen instances (e.g. *"white and blue box"*) and disambiguate objects based on spatial relations. Grounding failures are usually due to highly occluded objects in the scene, especially if multiple distractor objects are present. Several failure cases in cluttered scenes are due to collisions during motion execution, nevertheless, the 4-DoF grasp is correctly predicted. Detailed experiments are shown in the supplementary video.

## 6 Conclusion, Limitations and Future Work

This paper presents OCID-VLG, a new dataset for language-guided 4-DoF grasp synthesis in clutter, offering the first benchmark that connects language instructions with grasping in an end-to-end fashion. Further, we propose CROG, a CLIP-based end-to-end model as a solution. Extensive experimental comparisons and ablation studies validated the effectiveness of CROG over previous methods, and set a competitive baseline for our dataset. Overall, this research offers valuable insights into language-guided grasp synthesis and lays the foundation for future advancements in this field.

However, we found that CROG is limited when grounding concepts that lie outside the training distribution. We attribute this to the pretraining-finetuning strategy, which trades off the zero-shot capacity of CLIP pretraining in favor of the dense finetuning tasks. Future work will explore methods to efficiently learn the dense decoding tasks while maintaining better zero-shot grounding capability of CLIP. Finally, since CROG only considers RGB information, we would like to investigate whether further fusing depth data alongside RGB aids in grasp synthesis.

**Acknowledgements:** This work is supported by EU H2020 project "Enhancing Healthcare with Assistive Robotic Mobile Manipulation (HARMONY, 101017008)", as well as from Google DeepMind through the Research Scholar Program for the "Continual Robot Learning in Human-Centered Environments" project.

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

# A OCID-VLG Vocabulary

We visualize a word cloud of the concept vocabulary of OCID-VLG in Fig. 5, while the full attribute concept catalog is given if Fig. 6. Besides common sub-phrases such *"box", "food", "product"*, the wordcloud demonstrates that the most frequent concepts used to disambiguate objects are spatial predicates, both as pair-wise relations (*"front", "right", etc.*) and as absolute location (e.g. *"leftmost", "closest"*). Certain object names (e.g. *"kleenex", "tissues", "cereal"*) appear more frequently, as those are the objects that are most commonly ambiguous in OCID scenes, hence they spawn a lot of expressions referring to them. Finally, colors and brand names appear also frequently, as they are the most common discriminating attribute between objects of the same category.

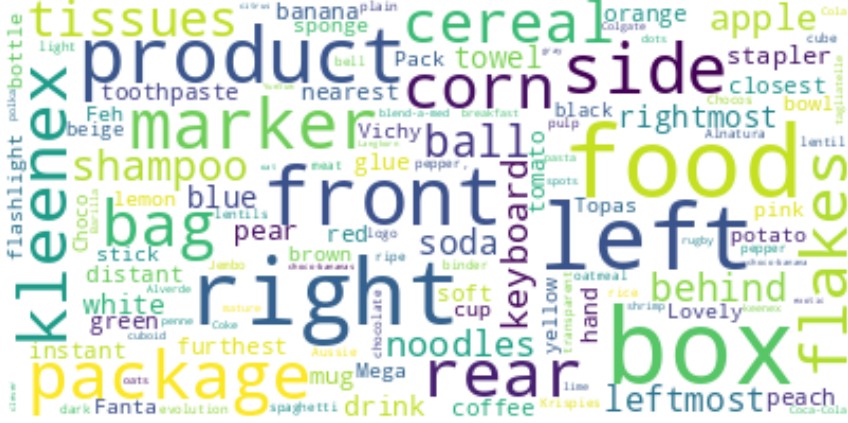

Figure 5: Wordcloud of OCID-VLG Vocabulary

The number of unique concepts per concept type, as well as the total number including paraphrases are presented in Table 5. Paraphrases include synonyms (e.g. *"Coca-Cola", "Coke"*) as well as different phrasings of relations (e.g. *"left of", "to the left side of"*).

| Concept | Num.Unique | Num.Total |
|---------|------------|-----------|
| Category | 30 | 55 |
| Color | 27 | 27 |
| Instance | 31 | 93 |
| Relation | 9 | 24 |
| Location | 4 | 8 |

Table 5: Number of concepts in OCID-VLG

Referring expressions might use instance-level names, attributes, relations, locations or combinations of all the above to disambiguate objects. We study the frequency of referring expressions on the OCID-VLG data splits in Table 6. Most frequent type is name (which includes a lot of variety

| Type | Train | Validation | Test |
|------|-------|------------|------|
| Name | 20678 | 3014 | 5809 |
| Attribute | 2739 | 348 | 781 |
| Relation | 20501 | 2792 | 5769 |
| Location | 9306 | 1285 | 2672 |
| Mixed | 9997 | 1230 | 2718 |

Table 6: Number of referring expressions in OCID-VLG organized by type

in concepts such as brand, flavor etc.) with pair-wise relations following closely. Spatial relations can always refer to the target uniquely by querying for a relation to a neighbouring object. Location

| | ID | class | label | color | material | special |
|---|---|---|---|---|---|---|
| 0 | 1 | apple | apple_1 | red | organic | NaN |
| 1 | 2 | apple | apple_2 | green | organic | NaN |
| 2 | 3 | ball | ball_1 | blue | plastic | NaN |
| 3 | 4 | ball | ball_2 | yellow | plastic | rugby ball |
| 4 | 5 | ball | ball_3 | red and white | plastic | polka ball,ball with spots,ball with dots |
| 5 | 6 | banana | banana_1 | yellow | organic | NaN |
| 6 | 7 | bell_pepper | bell_pepper_1 | red | organic | NaN |
| 7 | 8 | binder | binder_1 | green | plastic | NaN |
| 8 | 9 | bowl | bowl_1 | blue | ceramic | NaN |
| 9 | 10 | cereal_box | cereal_box_1 | red | paper | Topas box,Topas cereal box,Topas corn flakes,Topas cereal |
| 10 | 11 | cereal_box | cereal_box_3 | white and blue | paper | Mega Pack box,Mega Pack cereal box,Mega Pack cereal,Mega Pack corn flakes |
| 11 | 12 | cereal_box | cereal_box_4 | brown | paper | Choco Krispies box,Choco Krispies cereal box,Choco Krispies box,Choco Krispies corn flakes |
| 12 | 13 | cereal_box | cereal_box_5 | green and red | paper | Chocos box,Chocos cereal box,Chocos box,Chocos corn flakes |
| 13 | 14 | coffee_mug | coffee_mug_1 | black | ceramic | mug with evolution logo |
| 14 | 15 | coffee_mug | coffee_mug_2 | white | ceramic | plain mug |
| 15 | 16 | flashlight | flashlight_1 | black | metal | NaN |
| 16 | 17 | food_bag | food_bag_2 | transparent | plastic | lentil bag,bag with lentils |
| 17 | 18 | food_bag | food_bag_3 | red and white | plastic | pasta bag,penne bag,spaghetti bag,spaghetti penne bag,bag with pasta |
| 18 | 19 | food_bag | food_bag_4 | white | plastic | rice bag,Langkorn rice bag,clever rice bag,bag with rice |
| 19 | 20 | food_box | food_box_1 | dark blue | paper | Barilla box,tagliatelle,spaghetti box |
| 20 | 21 | food_box | food_box_2 | yellow and green | paper | chocolate banana box,choco-bananas,chocolate banana box,box with choco-banana |

Figure 6: Full attribute catalog of OCID-VLG

and mixed follow at about half frequency, while color is last, as several objects in OCID share color between different instances of the same category.

## B  Qualitative Results

We visualize predicted masks and grasp poses from the implemented baselines and the proposed CROG model in Fig. 7. We include two examples per referring expression type for test scenes of OCID-VLG dataset. Zero-shot baselines based on pretrained GR-ConvNet provide poor grasp proposals, while supervised baselines + CLIP (Det-Seg, SSG) are constrained by the ranking errors of CLIP. Due to segment-then-rank pipeline, spatial information about other objects is lost when considering only the mask of a single object. As a result, CLIP-based baselines struggles with grounding spatial relations. CROG is robust across referring expression types.

In Fig. 8, we visualize outputs of the CROG model during real robot experiments. The plots include predicted mask and grasp proposal, as well as the three decoded masks from CROG's grasp projectors (quality, angle and width masks). It should be noted that the corresponding input command is shown atop each image.

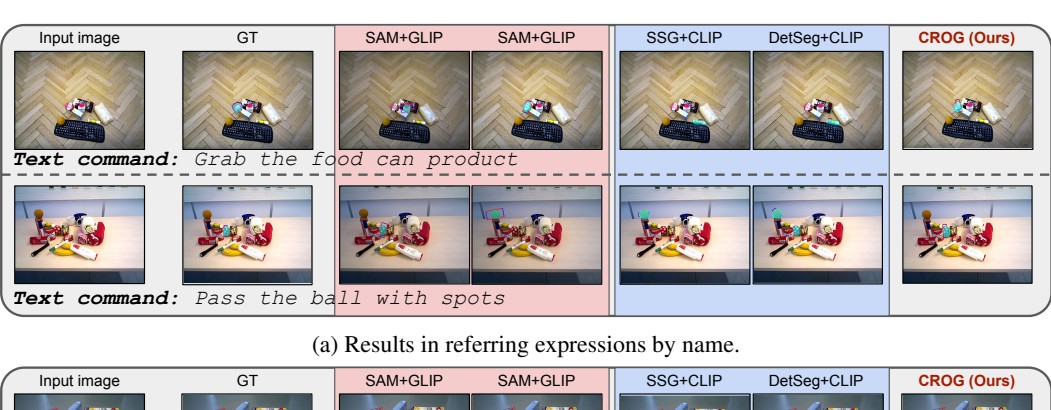

(a) Results in referring expressions by name.

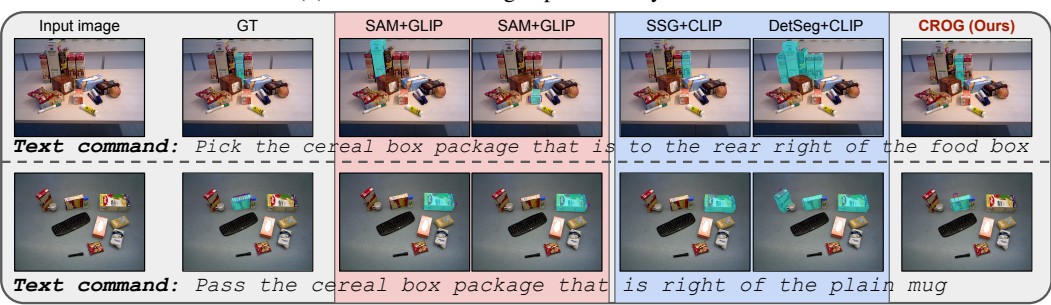

(b) Results in referring expressions by attribute.

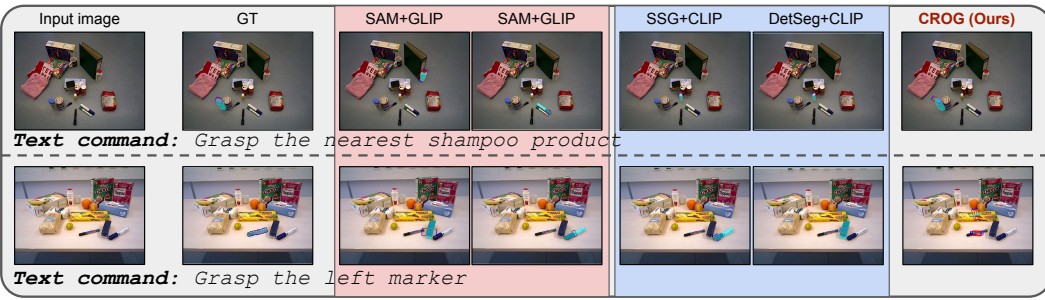

(c) Results in referring expressions by relation.

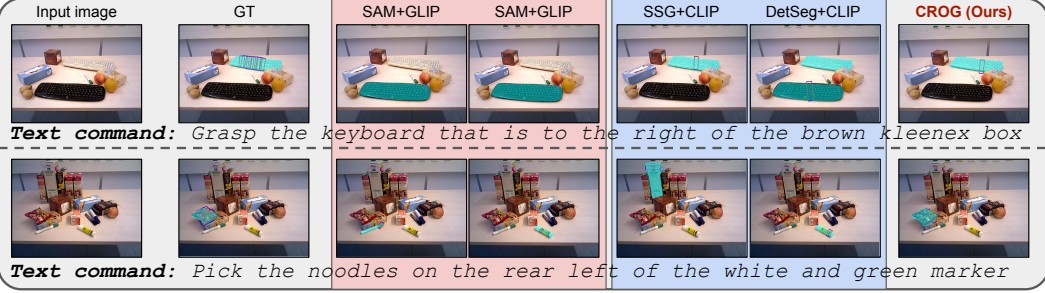

(d) Results in referring expressions by location.

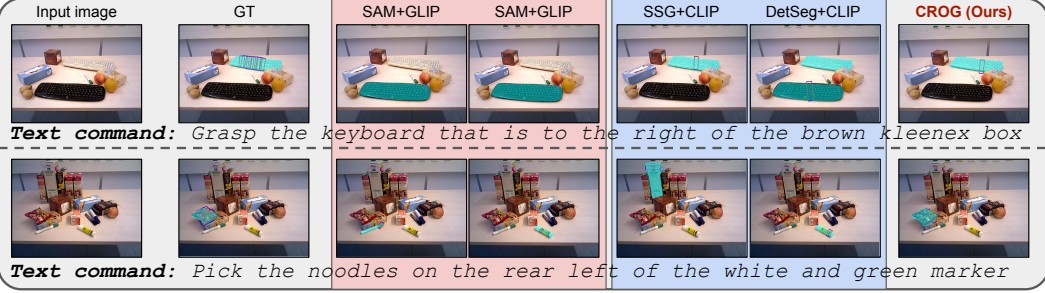

(e) Results in referring expressions by a mix of concepts.

Figure 7: Qualitative results in OCID-VLG test scenes.

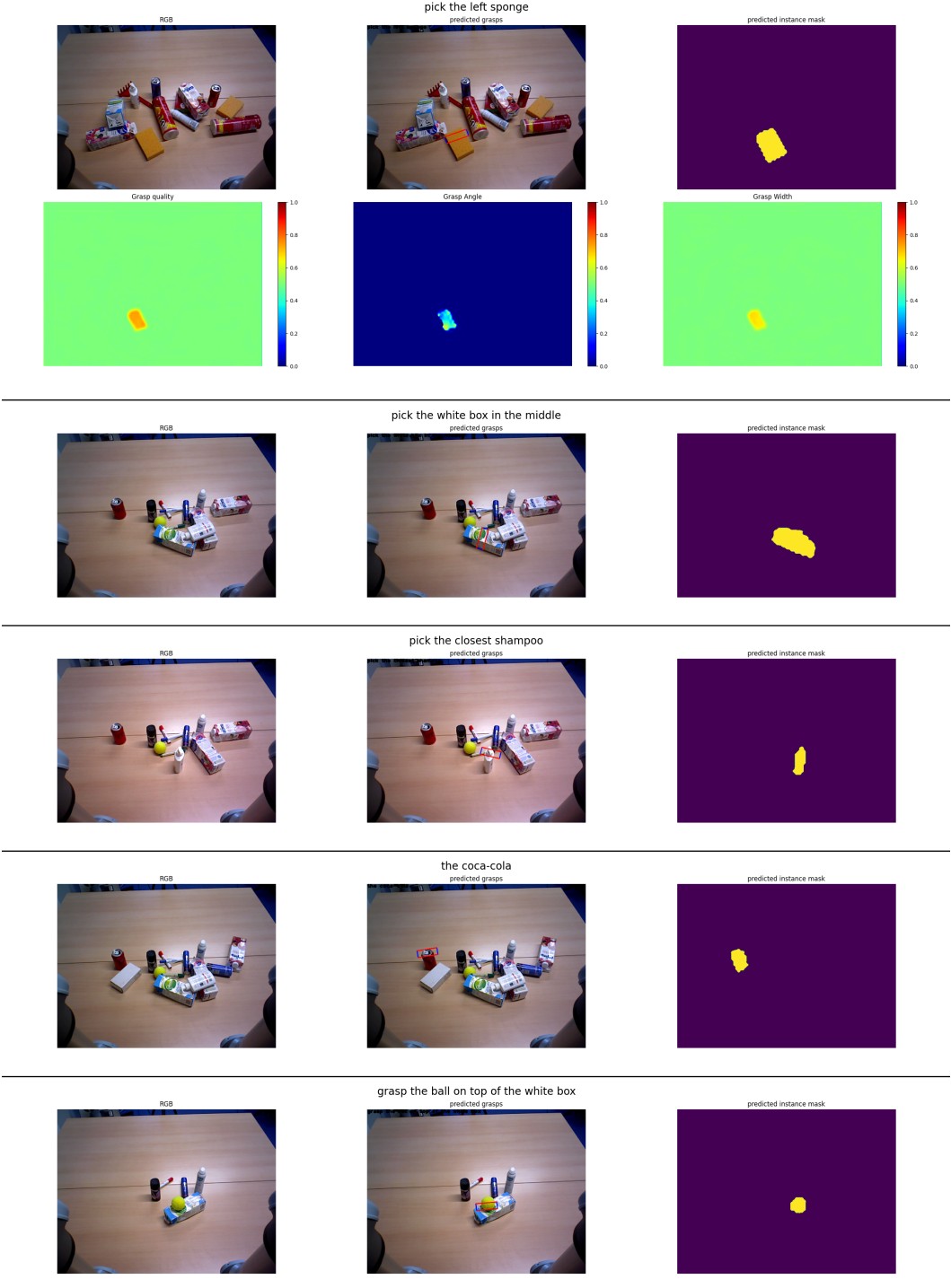

Figure 8: Qualitative results in real robot experiments.

