# OpenReview forum: "Language-guided Robot Grasping: CLIP-based Referring Grasp Synthesis in Clutter"
_robot-learning.org/CoRL/2023/Conference — CoRL 2023 Poster_

### Official Review · Reviewer_xdNw · 2023-07-05

**Confidence:** 4
**Originality:** Fair
**Technical Quality:** Good
**Clarity Of Presentation:** Very Good
**Impact:** 3

**Recommendation:**

Weak Accept: I recommend accepting the paper, but will not argue for my recommendation if the majority of other reviewers have a different opinion.

**Review:**

Strengths:
The new grasping dataset, OCID-VLG, based on OCID-Grasp, contains multi-modal information such as labeled grasps, vision data, and semantic parsing. If open-sourced, it could prove helpful to the research community for future open-vocabulary grasping work. The authors also clearly delineate the differences between this new dataset and existing ones.

Areas for Improvement:
The proposed method does not generalize well to objects outside the training distribution, which consists of 31 objects. Notably, many grasp synthesis papers [1,2] can generalize to a large distribution of objects, despite lacking semantic labels. The use of class-agnostic segmentation in combination with a grasp synthesis network [3] has demonstrated high success rates in real-world bin picking scenarios. The success rate for the proposed open-loop planner appears relatively low comparing to the aforementioned works.

The proposed method is a direct adaptation from CRIS. Broadly speaking, it accomplishes something very similar to CLIPort, which could be considered a single-stage grasp synthesis model. The key difference between the proposed approach and CLIPort, as the reviewer understands it, lies in how the CLIP model is used. While CLIPort uses a frozen CLIP model, the proposed approach fine-tunes CLIP. Reference [4] suggests that fine-tuning CLIP may lead to a reduction in its performance on long-tail queries. Such a comparison is absent in the manuscript, and the reviewer believes that conducting this experiment would help readers better understand the effect of CLIP in the proposed approach.

[1] Mahler, J., Liang, J., Niyaz, S., Laskey, M., Doan, R., Liu, X., ... & Goldberg, K. (2017). Dex-net 2.0: Deep learning to plan robust grasps with synthetic point clouds and analytic grasp metrics. arXiv preprint arXiv:1703.09312.
[2] Zhu, X., Wang, D., Biza, O., Su, G., Walters, R., & Platt, R. (2022). Sample efficient grasp learning using equivariant models. arXiv preprint arXiv:2202.09468.
[3] Danielczuk, M., Matl, M., Gupta, S., Li, A., Lee, A., Mahler, J., & Goldberg, K. (2018). Segmenting unknown 3D objects from real depth images using mask R-CNN trained on synthetic point clouds. arXiv preprint arXiv:1809.05825, 16.
[4] Kerr, J., Kim, C. M., Goldberg, K., Kanazawa, A., & Tancik, M. (2023). Lerf: Language embedded radiance fields. arXiv preprint arXiv:2303.09553.

**Quality Of The Limitations Section:**

Limitations are addressed clearly

**Questions For Rebuttal:**

1. Line 114-116 “The dataset consists of 1763 indoor tabletop RGB-D scenes with high clutter, including 31 object categories from a total of 58 unique instances.” Is the dataset generated based on a fixed camera pose or is there randomization for the camera poses?
2. For the physical experiment, has the authors experienced any difficulty when transferring the model that is trained purely with data generated in simulation to real?
3. Will the OCID-VLG dataset be open-sourced for public use?
4. Line 247: The proposed approach uses a fairly large input image dimension for CLIP (416x416 instead of the standard 224x224). What is the runtime of the pipeline (i.e. how long does it take to generate each grasp?)
5. Line 290: The CLIP baseline struggles with grounding spatial concepts and the proposed approach seems to have alleviated this problem. Is there any experiment on a table top setup where there are a few repeated objects and the only possible way to distinguish them is via spatial relation?

**Robotics Focus:**

Sufficient demonstration on hardware

**Summary Of Paper:**

This work focuses on robotic grasp synthesis with natural language object referral. In particular, many existing works utilize a multi-stage pipeline that separates the parsing of semantics and the synthesis of grasps. This study introduces a new benchmark for grasping in cluttered indoor scenes based on natural language expressions. Additionally, it presents a new single-stage, end-to-end model based on CLIP to generate grasps, demonstrating promising performance in spatial reasoning. The manuscript also includes tabletop 4DoF picking experiments conducted in both simulated and real environments.

**Summary Of Recommendation:**

Weak accept. The manuscript introduces a novel multi-modal grasping dataset that includes vision, language, and grasp data. Experiment results suggest that the proposed single-stage grasp algorithm outperforms two-stage algorithms. Some limitations: the dataset has a limited diversity of object choices, and does not yet support open vocabulary grasp queries.

---

### Official Review · Reviewer_upRx · 2023-07-21

**Confidence:** 4
**Originality:** Good
**Technical Quality:** Good
**Clarity Of Presentation:** Very Good
**Impact:** 3

**Recommendation:**

Weak Accept: I recommend accepting the paper, but will not argue for my recommendation if the majority of other reviewers have a different opinion.

**Review:**

Strengths
- I like the task of language gudied grasping, and appreciate the focus of the paper on referring expressions by spatial relations which is a notable lacking capability of many vision-language models
- The dataset is a useful platform for conducting vision-language grasping research

Weaknesses
- One of the main motivations of using pretrained vision-language models and language is to capture their internet-scale pretraining and language understanding capabilities. Fine-tuning on a small set of objects with limited appearance largely throws away this internet-scale pretraining, as evidenced by failure cases being "mostly for objects that are not similar in appearance to OCID-VLG categories", as the authors note. Authors acknowledge this limitation and leave it to future work, however I feel it makes the results less interesting given their limited scalability.
- The performance on grasping and grounding is not very high (around 20% and 60% in physical trials), and this is even for objects from similar categories as the train set (eg mug, marker, etc). I worry that CROG has essentially overfit to the provided dataset, raising questions about the difficulty of the dataset as well as the transferability of the method to practical deployment.
- Baselines are not tested on actual grasping performance, but just IoU with the dataset, which doesn't capture the full story of performance (indeed, this can be seen by IoU for CROG being very high, but physical success quite low). I feel the author's should have compared to a zero-shot baseline in physical trials, for example querying an off-the-shelf grasp planner like GraspNet-1B, DexNet, etc and masking the grasps by the segmentation mask from CRIS or SAM+CLIP. Spatial relations could be processed with a scene-graph type post-processing similar to the data collection method except operating on detected masks.

Comments
- I feel the connection to CRIS should be made apparent earlier in the introduction, given how heavily the method is based on it.
- It would be helpful for analysis to separate results by spatial relation and intrinsic properties of objects like color
- It would be helpful to provide an exhaustive list of all the object categories in the appendix (the word cloud is nice, but doesn't enable inspecting the difference between train/test object categories)


**Quality Of The Limitations Section:**

Limitations are addressed clearly

**Questions For Rebuttal:**

Questions
- Do held-out scenes in the OCID-VLG scenes contain seen objects? If objects in the test set were seen before but only in different configurations I feel this somewhat invalidates the point of a test set since it doesn't measure transfer to new objects, and should be made clearer in the text.
- In the video included of "ball on top of the box", the referring expression is redundant since there is only one ball in the scene. The text states "in each scene we include distractor objects of the same category as the queried object". Is this a mistake?
- Querying spatial relationships with 0-shot baselines is unfair since, as the authors note, the CLIP baselines have no access to spatial relationships given isolated object crops. It seems like this baseline is almost set up to fail, since including some rudimentary spatial post-processing would have been straightforward given a set of candidate objects, and ranking based on both CLIP similarity and spatial properties wouldn't be too challenging.
- It would strengthen the experiment section to include tests on objects which are completely outside the training categories (not just instances) to investigate how much language understanding is retained on unseen categories through finetuning.


**Robotics Focus:**

Sufficient demonstration on hardware

**Summary Of Paper:**

This paper addresses the task of grasping objects from language referring expressions which can include name, intrinsic properties, and spatial relationships to other objects. It proposes a relabeling scheme for an existing dataset to augment it with language annotations including disambiguating intrinsic and extrinsic properties. It further proposes a method adapted from prior work for training a grasp and segmentation network on the dataset which finetunes a CLIP model for segmentation and grasp prediction. It presents comparisons against zero-shot baselines using off-the-shelf segmentation and language models.

**Summary Of Recommendation:**

Overall I think the dataset and approach are valuable, so I weakly recommend accepting. However, I have some major concerns about the scalability of such a method, and about the evaluation protocol which I noted above.

---

### Official Review · Reviewer_ZgiR · 2023-07-22

**Confidence:** 5
**Originality:** Very Good
**Technical Quality:** Very Good
**Clarity Of Presentation:** Very Good
**Impact:** 3

**Recommendation:**

Strong Accept: I recommend accepting the paper and will argue for my recommendation even if other reviewers hold a different opinion.

**Review:**

Strengths:
- Grasping in the real world with language instructions is relatively unexplored, and this paper does a good job of benchmarking it.
- The use of the CLEVr engine to provide annotations for the dataset is clever.
- The proposed CROG method with joint training of segmentation and grasp quality masks is interesting and is shown to perform well
Extensive comparison and ablation experiments have been carried out.

Weaknesses:
- The proposed method may not work with any arbitrary language queries outside the dataset relation set.
- The authors could add a discussion about how language-based grasping could be applied to 6Dof grasping


**Quality Of The Limitations Section:**

Limitations are addressed clearly

**Questions For Rebuttal:**

- Can you hypothesize how you would use depth data in the pipeline? Where would it help the most?
- How can the method be extended to 6-DoF grasping problems?


**Robotics Focus:**

Sufficient demonstration on hardware

**Summary Of Paper:**

This paper introduces a dataset and method for 4-DoF grasping with language instructions, termed referring grasp synthesis. The dataset is based on the OCID dataset and adds language annotation to the grasps using the CLEVR engine. A novel method CROG is proposed, which is jointly trained on vision-language-based segmentation and grasp quality prediction. The method compares favorably to baselines and is demonstrated on a real dual-armed grasping setup.

**Summary Of Recommendation:**

The authors introduce a novel approach as well as a dataset that is very useful for top-down language-based grasping. Extensive experiments and ablations have been carried out with different handling of segmentation, language, and grasp synthesis modules. Sufficient real robot experiments have been carried out.

---

### Decision · Program_Chairs · 2023-08-30

**Decision:**

Accept (Poster)

**Comment:**

In overall, it is an interesting research paper. The task of language guidied grasping is new and useful for the community. The paper also introduces OCID-VLG, a novel and demanding dataset that explores complete vision-language-grasping scenarios within naturally cluttered environments. This dataset establishes a link with grasp annotations sourced from the OCID-Grasp dataset.

**Strengths**:

- An end-to-end CLIP-based model for pixel-level segmentation and grasp synthesis.

- Benchmarking of CLIP-integrated grasping methods in the real world with language instructions

- An interesting use of the CLEVr engine to provide annotations for the dataset

- The dataset is a useful platform for conducting vision-language grasping research

**Weaknesses**:

- The performance on grasping and grounding is not very high.

- Baselines are not tested on actual grasping performance, but just IoU with the dataset. The author's should have compared to a zero-shot baseline in physical trials.

- The dataset has a limited diversity of object choices, and does not yet support open vocabulary grasp queries.

- Generalization ability is limited

- Only with 4DoF grasp proposals

Post-rebuttal: Please revise the paper in line with the rebuttal discussion for the camera-ready submission. It would be useful for the community if the created dataset is published with the paper publication.